# Analysis of Astroglial Secretomic Profile in the Mecp2-Deficient Male Mouse Model of Rett Syndrome

**DOI:** 10.3390/ijms22094316

**Published:** 2021-04-21

**Authors:** Yann Ehinger, Valerie Matagne, Valérie Cunin, Emilie Borloz, Michel Seve, Sandrine Bourgoin-Voillard, Ana Borges-Correia, Laurent Villard, Jean-Christophe Roux

**Affiliations:** 1Aix Marseille University, Inserm, MMG, Marseille Medical Genetics, 13385 Marseille, France; Yann.Ehinger@ucsf.edu (Y.E.); Valerie.MATAGNE@univ-amu.fr (V.M.); emilie.BORLOZ@univ-amu.fr (E.B.); Ana.BORGES-CORREIA@univ-amu.fr (A.B.-C.); laurent.villard@univ-amu.fr (L.V.); 2Inserm U1055, LBFA and BEeSy, Prométhée Proteomics Platform, Université Grenoble Alpes, 38000 Grenoble, France; vcunin@chu-grenoble.fr (V.C.); MSeve@chu-grenoble.fr (M.S.); sandrine.bourgoin@univ-grenoble-alpes.fr (S.B.-V.); 3Prométhée Proteomics Platform, Institut de Biologie et de Pathologie, CHU Grenoble Alpes, 38000 Grenoble, France; 4University Grenoble Alpes, CNRS, Grenoble INP, CHU Grenoble Alpes, TIMC, PROMETHEE Proteomic Platform, 38000 Grenoble, France

**Keywords:** astrocytes, iTRAQ quantitative proteomic approach, Mecp2, neuronal arborization, Rett syndrome, secretome

## Abstract

Mutations in the X-linked *MECP2* gene are responsible for Rett syndrome (RTT), a severe neurological disorder. MECP2 is a transcriptional modulator that finely regulates the expression of many genes, specifically in the central nervous system. Several studies have functionally linked the loss of MECP2 in astrocytes to the appearance and progression of the RTT phenotype in a non-cell autonomous manner and mechanisms are still unknown. Here, we used primary astroglial cells from *Mecp2*-deficient (KO) pups to identify deregulated secreted proteins. Using a differential quantitative proteomic analysis, twenty-nine proteins have been identified and four were confirmed by Western blotting with new samples as significantly deregulated. To further verify the functional relevance of these proteins in RTT, we tested their effects on the dendritic morphology of primary cortical neurons from *Mecp2* KO mice that are known to display shorter dendritic processes. Using Sholl analysis, we found that incubation with Lcn2 or Lgals3 for 48 h was able to significantly increase the dendritic arborization of *Mecp2* KO neurons. To our knowledge, this study, through secretomic analysis, is the first to identify astroglial secreted proteins involved in the neuronal RTT phenotype in vitro, which could open new therapeutic avenues for the treatment of Rett syndrome.

## 1. Introduction

Rett syndrome (RTT) is a severe neurological disorder caused by mutations in the *MECP2* (Methyl CpG binding protein 2) gene, located on the X chromosome [1]. After a period of apparently normal development, females with *MECP2* mutations undergo a regression of early neurodevelopment, resulting in the deterioration of motor skills, purposeful use of the hands, speech, and the appearance of severe breathing disturbances as the disease progresses [2]. Mecp2 was originally described as a DNA methylation-dependent transcriptional repressor but has also been shown to play various roles in chromatin compaction and global gene expression [3]. Since Mecp2 is essentially expressed in neurons, where its expression correlates with neuronal maturation [4,5], and because it was thought to be absent in glial cells, RTT was considered to be the result of cell-autonomous neuronal defects.

Neurons have long been considered as the main players in the central nervous system (CNS), while glia was relegated to physical and metabolic support. However, over the past decade new research has highlighted the major roles played by glia in neural development, maintenance and functions [6,7,8]. Absence of Mecp2 in glia [9] has since been disproved [10,11,12,13,14] and astrocytes were finally implicated in RTT pathophysiology and described in many non-cell-autonomous mechanisms [10,11,15,16,17,18]. These studies suggested that abnormal astrocytes impair the normal neuronal function and participate in the progression of the disease. Conversely, the restoration of Mecp2 expression specifically in astrocytes of *Mecp2* KO mice delays the progression of RTT symptoms by improving locomotion, anxiety, breathing abnormalities and increasing their lifespan. Furthermore, re-expression of Mecp2 specifically in astrocytes leads to a non-cell-autonomous positive effect on *Mecp2* KO neurons in vivo, restoring normal dendritic morphology, which definitively implicates astrocytes in the pathophysiology of RTT [12].

In the last decade, several studies pointed out the key roles of secreted proteins (glial or non-glial) in neuronal functions in physiological and pathological conditions by acting on cell signaling, communication, transduction, angiogenesis and migration [19,20]. In the meantime, several efforts were done in mass spectrometry-based proteomic research (related to the field of secretomics) to identify and quantify these secreted proteins by minimizing the difficulties inherent to the high dilution factor of proteins and deciphering a deeper secretome, i.e., the set of proteins released from cells into the extracellular space [21]. Kim et al. also pointed out the interest of using proteomic studies to establish astroglial secretome profiles and determine the role of deregulated secreted proteins in neuronal disorders [20]. Interestingly, a study in an in vitro coculture system indicated that *Mecp2* KO astrocytes (or their conditioned medium alone) or human RTT astrocytes derived from IPSc have a harmful, non-cell-autonomous effect on wild-type (WT) neurons, resulting in a pathological neuronal phenotype and a short dendritic morphology [10,22]. We therefore hypothesized that the secretion or non-secretion of specific factors by *Mecp2* KO astroglia may lead to functional and morphological neuronal anomalies. In order to test this hypothesis, we analyzed the astroglial secretome through a quantitative mass spectrometry-based proteomic approach [19,23]. We evidenced several deregulated secreted proteins in the secretome of *Mecp2*-deficient astroglia compared to wild-type cells, providing new insights into RTT physiopathology.

## 2. Results

### 2.1. Proteomic Analysis of Astroglial Secretome

To determine which secreted proteins were deregulated in RTT, our quantitative proteomic analysis was performed on the astroglial secretome obtained from postnatal day (PND) 2 RTT male mice (*Mecp2* KO and WT, *n* = 6 per condition, split in two experimental replicates) coming from 2 independent experiments (Figure 1A). This led to the quantification of 557 proteins (Appendix A) by considering only proteins with at least 1 unique peptide with a confidence score higher than 95% and an FDR of 1%. 105 proteins (listed in Appendix A) were found as significantly deregulated in both replicates (with at least 2 unique peptides above 95% confidence level, a *p*-value ratio lower than 0.05) in the absence of Mecp2. The functional analysis performed using David Bioinformatics resources (https://david.ncifcrf.gov/summary.jsp, accessed on 2019) confirmed that the quantified proteins were mostly localized in extracellular compartments (extracellular exosome, extracellular space and extracellular region associated to a *p*-value of 2.7 × 10–47, 2.9 × 10–15 and 3.1 × 10–13, respectively) (Appendix A). In addition to these Gene Ontology (GO) annotations, the Uniprot annotation provides subcellular localization of proteins with both expertly curated data from literature and rule-based automatic annotation. Therefore, we further filtered the list of our deregulated proteins by using the subcellular localisation of Uniprot annotation. Altogether, by considering GO terms and Uniprot annotations, twenty-nine secreted deregulated proteins were retained (Table 1 and Appendix A) to build a network of secreted deregulated proteins upon Mecp2 deficiency in the RTT astroglial secretome (Figure 1C). Among these 29 secreted proteins, 11 were down-regulated while 18 proteins were up-regulated. The heat map of deregulated secreted proteins (Figure 1B) confirmed the coherence of the iTRAQ ratio (*Mecp2* KO vs WT) obtained from the two independent replicates with the highest fold change for Lcn2 (iTRAQ ratio KO/WT 0.31), Lgals1 (iTRAQ ratio KO/WT FG 0.35), Mfap4 (iTRAQ ratio KO/WT FG 2.83), and Enpp5 (iTRAQ ratio KO/WT FG 2.54) (Table 1). Although our analysis highlighted a small number of proteins for global profiling (29 deregulated secreted proteins), this study provides the first proteomic insight into astroglial-secreted factors in RTT. The secretomic data were deposited to the ProteomeXchange Consortium via the MassIVE Dataset Submission [24] under the MassIVE MSV000085665 identifier (http://massive.ucsd.edu) and ProteomeXchange PXD020143 identifier (http://www.proteomexchange.org).

### 2.2. Functional and Network Analyses of Secreted Proteins Deregulated upon Mecp2 Deficiency

The STRING network of the secreted proteins significantly deregulated in the RTT astroglial secretome is shown in Figure 1C and Appendix A. The highest betweenness centrality scores are obtained for Lcn2 and Lgals3, found downregulated in the RTT astroglial secretome. Knowing that betweenness centrality reflects the power proteins exert on the interactions of other proteins (nodes) in the network, this suggests Lcn2 and Lgals3 as the most pivotal proteins of the network. Based on the functional analysis (Figure 1D and Appendix A), the up-regulated proteins in the RTT astroglial secretome are mostly involved in inflammatory response, positive regulation of neuron death, complement activation, metabolic process, cellular response to UV-B, and negative regulation of beta-amyloid formation. The down-regulated proteins are mostly involved in the apoptotic process, maintenance of protein location cellular response to interleukin-1, and response to drug.

### 2.3. Validation of Proteomic Data by Western Blot

Among the 29 deregulated proteins validated by mass-spectrometry analysis, we analyzed by Western blotting the four most relevant proteins on a translational level including both proteins (Lcn2 and Lgals3) with the highest betweenness centrality scores in the network of deregulated secreted proteins. We quantified protein level for Lcn2, Lgals3, B2M and Mfap4 (respectively −45%, −37%, −25%, and +55% of expression level by mass spectrometry in *Mecp2* KO secretome compared to WT) in three new secretome samples of astroglial culture from WT and *Mecp2* KO mice. Significant downregulation of Lcn2 (−42%), Lgals3 (−44%) and a trend of downregulation of B2M (−43.4%) was confirmed in the *Mecp2* KO astroglial secretome samples by Western blot (Figure 2 and Appendix A). Although secretomic analysis showed an important upregulation of Mfap4, we only found a trend to an upregulation of Mfap4 (Figure 2). All Western blots were normalized with total protein staining (Figure 2).

We also analyzed protein expression of these factors in astroglia culture. As Lcn2 is a secretory protein, its cellular levels were too low to be analyzed by Western blot (data not shown). Interestingly, no significant differences in Lgals3, B2M, and Mfap4 protein levels were found between WT and *Mecp2* KO astroglial cells (Appendix A), which suggests that the observed astroglial secretome deregulations could be the consequence of secretion rather than protein synthesis deficit.

### 2.4. Lcn2 and Lgals3 Supplementation Improves Dendritic Arborization of Mecp2 KO Neurons

An important feature of RTT is the reduced dendritic arborization that can be found in both RTT mouse models and RTT patients [25,26] and was also shown in this study (Figure 3A). Astrocytes are known to participate in neuronal dendritic growth [27]. To correlate the observed deregulations in the astroglial secretome with RTT features, we tested the effect of two deregulated factors on neuronal dendritic morphology. We focused on Lcn2 and Lgals3 for several reasons: (1) Lcn2 mRNA has been already found deregulated in *Mecp2* KO astrocytes coming from a mouse model of RTT [17], (2) Lcn2 has been shown to modulate spine morphology and to regulate neuronal excitability [28,29] and (3) Lgals3 is strongly expressed in the brain and modulates gliogenesis and memory [30,31]. Therefore, we supplemented *Mecp2* KO primary cortical neurons for 48 h with these factors and analyzed their dendritic morphology by using Sholl analysis. We also supplemented cortical neurons with NGF as a positive control of neurite outgrowth [32], because it leads to an increased branching and an increased total dendritic length in both WT and *Mecp2* KO neurons (Figure 3 and Appendix A). Sholl analysis showed that Lcn2 treatment significantly increased dendrite branching at 15–50 µm from the soma in *Mecp2* KO neurons (Figure 3). Similarly, Lgals3 treatment promoted dendrite branching at 20–60 µm from the soma in *Mecp2* KO neurons (Figure 3B). As shown in Appendix A, Lcn2 and Lgals3 treatment significantly increased dendrite branching in WT neurons as well.

## 3. Discussion

Astrocytes secrete various substances, including neuromodulators, classical neurotransmitters, growth factors, metabolic mediators, inflammatory factors, molecular regulators of synaptogenesis and synaptic connectivity factors [33]. It is well-documented that astrocytes and astrocyte conditioned medium are involved in RTT neuronal phenotype in vitro [10,11]. In the present study, we report significant changes in the RTT astroglial secretome and identify actionable targets. We first characterized the astroglial secretome in the absence of Mecp2 and we further identified and selected deregulated factors, then verified their relevance as potential therapeutic factors in an in vitro model.

### 3.1. Limitations

Despite the difficulties inherent to the high dilution factor of secreted proteins, we obtained reliable quantitative data that enabled us to identify and quantify 557 secreted proteins that represent the main biological pathways affected by RTT.

Astroglial cells were isolated from cortical regions and the deregulated secreted proteins, that have been identified and verified, may be specific to this region. Indeed, recent work from several labs reported the existence of morphological and transcriptional differences in astrocyte populations depending on their brain region [34].

Another limitation is the purity of our astrocyte population. Our purification protocol for cortical astrocytes does not yield pure astrocytes but rather a population highly concentrated in astrocytes with a low percentage of microglial cells, which means that microglial dysfunction may have contributed to our results. Precise astrocyte enrichment has not been assayed. However, proteins found deregulated are fully from astroglial origin since cultures were found free from neurons (Appendix A).

The impact of microglia cannot be neglected because, for instance, in the CNS, the Lcn2 protein is not only secreted by astrocytes and endothelial cells [35,36,37], but also by microglia [38,39]. Likewise, Lgals3 has also been identified as secreted by microglia [40,41].

### 3.2. GO Analysis Reveals Enriched Biological ProcessesiIn the RTT Astroglial Secretome

Despite the above-mentioned limitations, we identified 557 proteins, including conventionally secreted and non-conventionally secreted proteins as well as cytosolic proteins. Hence, 105 proteins were significantly deregulated in the *Mecp2* KO secretome compared to the WT one, in two independent biological replicates. Among these proteins, we focused our analysis on 29 proteins reported as secreted in the UniprotKB/Swissprot database. Although our analysis highlighted a small number of proteins for global profiling (29 deregulated secreted proteins), this study provides the first proteomic insight into astroglial-secreted factors in RTT. Further analysis based on GO terms revealed a neuro-inflammation signature in the RTT astroglial secretome with the up-regulation of proteins involved in the inflammatory response (C3, C4b, Hyal1) and down-regulation of proteins involved in eosinophil chemotaxis and cellular response to interleukin (Ccl2, Lgals3, Chil1, Lcn2). Our results also reveal the activation of the complement cascade through C3 and C4, important for the regulation of immune processes. These data are consistent with the fact that RTT is associated with neuroinflammation mechanisms and microglial activation [42,43,44]. Interestingly, a down-regulation of pro-apoptotic proteins in the secretome of glial cells (Chil1, Cst3, Lgals1, Lcn2) was observed and corroborates the view that Mecp2 deficiency reduces apoptosis-triggered senescence in neurons [45].

### 3.3. Network Analysis Identifies Two Proteins of Interest: Lcn2 and Lgals3

The highest betweenness centrality scores of the STRING secreted network (Figure 1C and Appendix A) upon Mecp2 deficiency were obtained for lipocalin-2 (Lcn2) and galectin-3 (Lgals3). These proteins observed as downregulated in the RTT astroglial secretome are involved in biological processes (eosinophil chemotaxis, apoptosis, maintenance of protein location, tumor necrosis factor, response to interleukin and several responses to drug) that are themselves known to play a key role in the modulation of neuronal functions in physiological and pathological conditions [46,47,48].

Moreover, a network analysis exploring the connections between Lcn2, Lgals3 and Mecp2 (Appendix A) showed that the known interactors are involved in the regulation of ERK1 and ERK2 cascade (Pten, Tlr4, Tlr2), positive or negative apoptotic/necrosis process (Pten, Fn1 Tlr2, Tlr4, Mmp9), metabolic process (Gapdh), regulation of cytokine and interleukin production (Tlr4, Tlr2), and other inflammatory processes (Pten, Mmp9). Interestingly, the Ras/ERK pathways mediated by IGF-1 and BDNF were already pointed out as playing a key role in RTT [49].

Therefore, we have selected these two secreted factors, lipocalin-2 (Lcn2) and galectin-3 (Lgals3) for further functional investigations.

### 3.4. A Well-Known Candidate, Lcn2

Lcn2 has been associated with diverse cellular processes in the CNS such as migration, cell death/survival, morphology and phenotypic polarization [39,50,51,52,53,54]. Our results showed a downregulation of Lcn2 in the RTT astroglial secretome. Interestingly, Lcn2 has already been identified as deregulated in astrocytes from two different RTT mouse models [17,55], in total brain samples from *Mecp2*-null mice [56] and in postmortem brains from RTT patients [57]. Lcn2 has been shown to modulate spine morphology and to regulate neuronal excitability, two parameters that are largely deregulated in RTT [28,58]. In addition, Lcn2-null mice display synaptic impairment in hippocampal long-term potentiation [28]. Therefore, a lack of Lcn2 secretion by *Mecp2* deficent astrocytes may be involved in spine morphology alteration and neuronal excitability defects in neurons. Indeed, we demonstrated in the present study that in vitro recombinant Lcn2 treatment increased neuronal arborization of *Mecp2*-deficent neurons. Further experiments are warranted to investigate the role of Lcn2 in the RTT neuropathology in vitro and, ultimately, *in vivo*.

### 3.5. A Multifunction Protein, Lgals3

Lgals3 acts as an endogenous ligand of TLR4 [40]. Lgals3 is known to be involved in the inflammatory response, and its expression is increased in microglia upon various neuroinflammatory stimuli [41,59]. Interestingly, deregulated inflammatory and immune responses have been found in RTT [60]. Moreover, it has recently been reported that Lgals3 regulates memory formation [61], and during development, Lgals3 has a critical role in driving oligodendrocytes differentiation, myelination, and gliogenesis [30,62].

### 3.6. Lcn2, Lgals3 and BDNF

Lcn2 and Lgals3 seem both involved in microglia and astrocyte activation [39,40]. Furthermore, conditioned media from Lcn2-activated astrocytes, and microglia upregulates pre- and postsynaptic markers such as synaptophysin, synaptotagmin and PSD95 in neural cultures. In addition, Lgals1 and Lgals3 have also been implicated in BDNF regulation [63]. Interestingly, our proteomic results suggest a downregulation of both Lgals1 et Lgals3 in the RTT astroglial secretome.

Given the essential role of BDNF in RTT physiopathology [64,65], Lgals1, Lgals3 and Lcn2 astroglial deficiency could be one of the numerous mechanisms leading to BDNF downregulation in this pathology. In agreement with this hypothesis, an in vitro study showed that Lcn2 treatment of rat astrocytes induced BDNF synthesis and release [66], while Lgals1 KO induces a decrease of BDNF levels in the prefrontal cortex [63] and Lgals3 deficient mice showed lower *bdnf* mRNA and protein levels in the hippocampus [67].

Based on these findings, we cannot exclude that the arborization improvement observed in *Mecp2* KO neurons treated with either recombinant Lgals3 or Lcn2 are BDNF-dependent.

## 4. Materials and Methods

### 4.1. Animals

The *Mecp2*-deficient mice (B6,129P2(C)-*Mecp2* tm1-1Bird) were obtained from the Jackson Laboratory (Charles River laboratories, Chatillon-sur-Chalaronne, France) and maintained on a C57BL/6 background. Genotyping was performed by PCR-amplification following a previously described protocol (Miralvès et al., 2007). The animals were housed under a 12:12 h light⁄dark cycle (lights on at 07:00) and given free access to food and water. The experimental procedures were carried out in keeping with the European guidelines for the care and use of laboratory animals (EU directive 2010/63/EU), the guide for the care and use of the laboratory animals of the French national institute for science and health (INSERM). Experimental protocols were approved by the ethical committee of the Aix Marseille University and the French M.E.N.E.S.R. minister (Permit Number: 02910.02 from 1 July 2016 to 30 June 2021 and 220319 from 07 June 2019 to 06 June 2024). All experiments were designed to minimize animal suffering.

### 4.2. Reagents

All culture media, supplements and material were purchase from Life Technologies (ThermoFisher Scientific, Courtaboeuf CEDEX, France) unless specified otherwise.

### 4.3. Primary Astroglial Culture and Medium Collection

On PND 2, male pups were rapidly decapitated, and the cortices were dissected in ice-cold sterile DPBS without calcium and magnesium under a binocular microscope. A tail biopsy was also taken and processed for genotyping. Cortical samples from each mouse were processed individually. Samples were washed twice in DPBS and transferred to culture media (DMEM-F12 medium with 10% fetal bovine serum (FBS) and 1X penicillin/streptomycin or pen/strep). Tissues were then cut in small pieces with a scalpel and dissociated by passing five times through a 21 g needle. Cell suspensions were then filtered through a 70-µm cell strainer and pelleted by a 10-min centrifugation at 200× *g*. Cell pellets were subsequently resuspended in culture media and plated in T25 culture flasks (Dutscher, Brumath, France) coated with poly-D-lysine solution (#P7886, 0.0001%, Sigma Aldrich, Saint-Quentin Fallavier, France). The medium was changed 24 h after plating and then thrice a week. Once confluency was reached, cells were trypsinized by incubating 5 min in phenol red free Tryple express solution, washed, centrifugated at 200× *g* for 10 min and plated on 10 cm culture dish (Falcon, Dutscher). When cells reached confluency, they were thoroughly washed with DPBS and the medium was replaced by serum free DMEM (#A1443001) with glucose (1 g/L), NaPyruvate (1 mM), 1X Pen/Strep, and without phenol red, glutamine or HEPES. Half the medium was then replaced by fresh medium every 4 days for 16 days. The collected medium was centrifuged at 200× *g* to pellet any cells or cellular debris and stored at −20 °C until needed. At the last medium collection, cells were washed three time in DPBS and stored at −85 °C until needed.

### 4.4. Secretomics

#### 4.4.1. Sample Purification

Once all medium samples were collected, they were thawed and passed through a 0.22 µm PTFE filter (Agilent Technologies, Montpellier, France). The medium was concentrated by centrifugation at 4000× *g* through Amicon Ultra-15 filters (PLBC, membrane Ultracel-PL, 3 kD, Merck-Millipore, St Quentin en Yvelines Cedex, France). Once all medium was filtered, the columns were rinsed three time with 10 mL of ultrapure water (Gibco), followed by a 20 min centrifugation at 4000× *g*. All samples from the same culture and from the same genotype were then pooled and concentrated by SpeedVac for 60 min at room temperature.

#### 4.4.2. Protein Digestion and iTRAQ Labelling

Sixty milligrams of secreted proteins per sample were digested and labeled with isobaric tagging reagents according to the manufacture’s recommendations (ITRAQ Reagents 4 plex Application kit; Sciex, Framingham, MA, USA). Briefly, samples were dried in a vacuum concentrator, dissolved in 0.5 M TEAB, reduced (50 mM TCEP), alkylated (200 mM TCEP), and digested by adding the trypsin/LysC Mix (Promega, Madison, WI, USA). Next, samples were labeled using iTRAQ reporter ion tags for 2 h, as follows: WT1 for *m*/*z* 114 tag, WT2 for *m*/*z* 115 tag, KO1 for *m*/*z* 116 tag, and KO2 for *m*/*z* 117 tag.

#### 4.4.3. Peptide Separation by OFFGEL Isoelectrofocusing and Nano LC-MS/MS Analysis

Samples containing iTRAQ labeled peptides were then pooled and fractionated in two steps as previously described [23,68]. First, peptide fractionation according to their pI by OFFGEL 3100 fractionator system using OFFGEL kit linear pH 3–10 (Agilent technology) in a 24-well configuration was performed. Peptides were dried, suspended in OFFGEL buffer and loaded into each well. Peptides were focused in a constant current of 50 µA until 50 kVh was accumulated. After isoelectrofocusing, the 24 OFFGEL fractions were desalted with C18 Zip-Tips (Millipore, Burlington, MA, USA) prior a fractionation with nano-HPLC using an Ultimate 3000 C18 RP-nanoLC system (Ultimate3000, Dionex/Thermo Scientific) controlled by Chromeleon v.6.8 software (Dionex/Thermo Scientific/LC Packings, Amsterdam, The Netherlands), and coupled to a Probot MALDI spotting device controlled by the µCarrier 2.0 software (Dionex/Thermo Scientific/LC Packings, Amsterdam, The Netherlands). Peptides were loaded on a trapping column (C18, 3 µm, 100 Å pore size; Thermo Scientific) in 2% ACN and 0.05%TFA for 5 min at a flow rate of 20 µL/min before introducing them into the reverse phase nano-column (Acclaim PepMap 100, 75 µm, 15 cm, nano-viper C18.3 µm, 100 Å pore size; Thermo Scientific) and separating them thanks to a binary gradient of buffer A (2% ACN, 0.05% TFA) and buffer B (80% ACN, 0.04% TFA) at a flow rate of 0.3 µL/min. The gradient was designed as follows: 0–5 min, 4% B; 5–35 min, 8%–42% B, 35–40 min, 42%–58% B; 40–50 min, 58%–90% B and 50–60 min, 4% B. Eluted fractions were automatically mixed with alpha-cyano-4-hydroxycinnamic acid matrix (HCCA, 2 mg/mL in 70% ACN and 0.1% TFA) at a ratio 1:3 v:v and spotted on an Opti-TOF LC/MALDI Insert 123*81 mm plate (Applied Biosystems, Foster City, CA, USA) at a frequency of one spot per 15 s.

Afterwards, spotted peptides were analyzed in MS and MS/MS modes using a 4800 MALDI-TOF/TOF mass spectrometer (Sciex, Les Ulis, France) operating in a positive reflector ion mode. The instrument was controlled by the 4000 Series Explorer software v. 3.5.3. Peptide Calibration Standard II (Bruker Daltonics, Bremen, Germany) was used to perform the external calibration with a mass tolerance of 50 ppm. MS spectra were acquired for each spot from 700 to 3500 *m*/*z*. The 30 most intense ions in each MS spectrum above a S/N (Signal/Noise) threshold of 30 were selected for MS/MS analysis. The fragmentation of the precursor ions was induced by using CID (Collision-Induced Dissociation) activation mode.

#### 4.4.4. iTRAQ Data Analysis

MS and MS/MS spectra were used for identification and relative quantification using the ProteinPilot software v 4.0 (Sciex, Les Ulis France) and the Paragon^TM^ algorithm. The analysis was performed with the Mus musculus UniProtKB/Swiss-Prot database (European Bioinformatics Institute, Hinxton, UK). The Paragon searches were run using thorough ID and a false discovery rate (FDR) of 1%. Only proteins with at least one peptide above 95% confidence level were included for the quantification analysis. As described previously [23,68,69], the statistical analysis was performed by implementing the area of the iTRAQ reporter ions for each of these validated peptides into the R Package Isobar v. 1.14.0. [70]. A protein was considered as significantly deregulated when pValueRatio was obtained lower than 0.05 in both replicates and at least unique 2 peptides were quantified.

In other words, we targeted our study on proteins recorded with the “secreted” keyword in UniProtKB entries that refers to proteins located outside the cell membrane without external protective or external encapsulating structures (i.e., space outside of plasma membrane and vesicles) after a published literature curation.

#### 4.4.5. Gene Ontology Analysis

A gene ontology analysis based on DAVID v.6.8 Gene Functional Annotation Tool available at https://david.ncifcrf.gov/home.jsp (accessed on 2019) [71,72] was performed on (i) identified proteins to check their localization and determine whether or not they were secreted proteins and on (ii) significantly deregulated secreted proteins to determine in which biological processes these proteins were involved. Only the proteins recorded with the “secreted” keyword in UniProtKB entries (https://www.uniprot.org/) (Uniprot, 2019), that refers to proteins located outside the cell membrane without external protective or external encapsulating structures (i.e., space outside of plasma membrane and vesicles) after a published literature curation, were targeted. A protein–protein interaction analysis of the secreted proteins significantly deregulated in *Mecp2* KO astroglial secretome was performed by using Cytoscape (version 3.8.0, Cytoscape Consortium, Chesterfield, MO, USA, https://cytoscape.org/) Shannon et al., 2003 with STRINGApp and STRING database (http://string-db.org, accessed on 2019) [73].

### 4.5. Western Blotting

Conditioned medium from 3 animals of the same genotype were pooled and then proteins were extracted and concentrated as described above (cf. 2.5.1). Proteins from astroglial culture were extracted with lysis buffer containing 50 mM Tris–HCl (pH = 7.5), 150 mM NaCl, 2 mM EGTA, 2 mM EDTA, 1% Triton X-100, 10 nM betaglycerophosphate, 5 mM Sodium pyrophosphate, 50 mM sodium fluoride and Halt^TM^ proteases and phosphatases inhibitor cocktail (Pierce Thermofisher). Protein concentrations of astroglial culture or concentrated medium were determined by the bicinchoninic acid method. Proteins extracted from conditioned medium of 3 animals of the same genotype were pooled. After a denaturation step at 96 °C for 5 min, proteins (100 μg) were separated on 4–20% SDS-polyacrylamide gel (Life Technology) and transferred onto a nitrocellulose membrane by electroblotting using the Trans-Blot turbo transfer system (Bio-Rad). The membrane was blocked with blocking buffer (Millipore, WBAVDFL01) for 1 h at room temperature. Primary antibodies for Lcn2 (1:1000, rabbit, abcam Ab63929), Lgals3 (1:1000, rabbit, abcam Ab76245), B2M (1:500, rabbit, Santa Cruz SC8362), Mfap4 (1:1000, rabbit, abcam ab169757), tubulin (1:5000, mouse, Sigma) and Actin (1:10000, mouse, Millipore), were diluted in the same solution and incubated overnight at 4 °C. The membrane was incubated using appropriate fluorescent secondary antibodies (IRDye 800 CW, IRDye 680 RD, LI-COR and LI-COR odyssey Imager). Quantitative analyses of signal intensity were performed using ImageJ software. For quantification in secretome samples, revert total protein stain (LI-COR Biotechnology, Bad Homburg, Germany) was used to normalize total protein quantity following the manufacturer’s recommendations.

### 4.6. Primary Neuronal Cultures and Treatments

On PND 7, male pups were rapidly decapitated, and the cortices were dissected in ice-cold sterile DPBS without calcium and magnesium and 1 g/L glucose under a binocular microscope. A tail biopsy was also taken and processed for genotyping. Each cortical sample from a mouse was processed individually. Primary cortical neurons were obtained by enzymatic digestion using the papain dissociation kit (Worthington, # LK003150, Serlabo Technologies, Vedene Cedex, France) according to the manufacturer’s recommendations. Briefly, cortices from one animal were minced by scalpel and incubated in the digestion solution (20 units/mL papain and 0.005% DNase in EBSS or Earle’s Balanced Salt Solution) at 37 °C in a rotating incubator for 45 min. The samples were dissociated by pipetting up and down 5 times with 1 mL pipet tip twice and then centrifuged at 300× *g* for 5 min at room temperature. The cell pellets were resuspended and separated by density gradient using the albumin-inhibitor solution provided. After a centrifugation at 70× *g* for 6 min at room temperature, the supernatant was discarded and the cell pellet immediately resuspended in culture medium (Neurobasal A medium, B27 1X, Glutamax 0.5 mM, Pen/Strep 1X). Cell suspension was filtered through a 70 µm cell strainer, centrifugated for 10 min at 100× *g* and resuspended in culture medium. After counting, cells were plated at a density of 40,000 cells/well on poly-d-Lysine-coated coverslips (1 µg/mL) in 24 well plate. After 2 h, the medium was replaced by culture medium with NGF (10 ng/mL) for 24 h. Forty-eight hours after plating, half the medium was replaced with fresh medium without growth factor or with either NGF (control), lipocalin 2 (50 ng/mL final concentration, #1857-CC-050, Bio-techne) or galectin 3 (1 µg/mL, final concentration, #9039-GA-050, Bio-techne). After 48 h of incubation, cells were then rinsed in DPBS and fixed by a 10-min incubation in 4% paraformaldehyde-PBS solution.

### 4.7. Immunofluorescence

Coverslips were washed three times in PBS buffer for 5 min, incubated in blocking solution (3% normal goat serum (Jackson ImmunoResearch Europe, Ltd., Suffolk, UK), 0.2% Triton X-100, PBS 1X) for 45 min and transferred in primary antibody (β3-Tubulin (D71G9) XP^®^ Rabbit mAb, #5568, 1:1000; Cell Signaling Technology, Ozyme, Saint Quentin Yvelines CEDEX, France) diluted in AB solution (3% normal goat serum, 0.1% Triton X-100, PBS 1X) or in AB solution only for 24 h at 4 °C. After washing 3 times for 5 min, coverslips were incubated for 1 h at room temperature with the secondary antibody (anti-goat Alexa 488, #A11055, ThermoFisher Scientific) diluted at 1:1000 in AB solution and then washed three more times in PBS. Coverslips were then air-dried for 30 min and mounted on glass slides with Shandon Immu-Mount medium (ThermoFisher Scientific).

### 4.8. Sholl Aanalysis

Pictures of isolated neurons were taken at 40X magnification on a Zeiss Apotome microscope (Carl Zeiss Microimaging, Jena, Germany). Exposition times, images processing and merging were done using the same parameters for each experiment.

Using the Fiji software [74], fluorescence images were converted to 8-bit format, then thresholded using the Li filter, background noise was removed and the Sholl analysis tool (default parameters, v3.7.3) was used to count the number of intersections between neuronal processes and 5 µm concentric circles [75].

### 4.9. Statistical Analysis

Statistical analysis on proteomic data was performed by using the R Package Isobar v. 1.14.0 [70] as already reported [23,68,69]. All other statistical analyses were performed using GraphPad Prism for Windows/MacOS (GraphPad Software, La Jolla, CA, USA, http://www.graphpad.com).

## 5. Conclusions

To our knowledge, this in vitro study is the first to identify secreted proteins that are deregulated in the *Mecp2* KO astroglia and that could be responsible for the neuronal RTT phenotype. The identification of proteins abnormally secreted by RTT astroglia, in addition to contributing to the RTT phenotype, could potentially lead to the development of new therapeutic strategies.

## Figures and Tables

**Figure 1 ijms-22-04316-f001:**
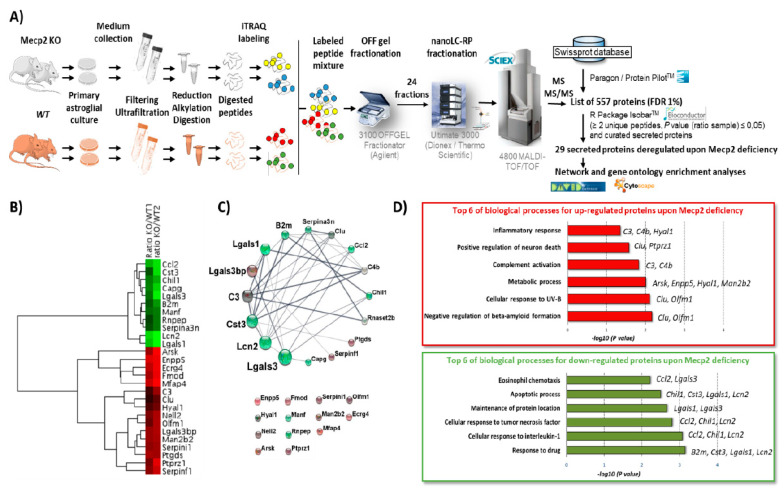
(**A**) General workflow for the iTRAQ labeled quantitative proteomics of glial secretome in the context of Rett syndrome (**B**) Heat Map of secreted proteins significantly dysregulated in the glial secretome of mouse Rett syndrome under hierarchical clustering (pairwise average-linkage/Euclidean distance of iTRAQ ratio KO/WT for both mice) (Cytoscape v 3.8.0 and MarkerCluster2) (**C**) STRING network of secreted proteins significantly dysregulated in glial secretome of mouse Rett syndrome (Cytoscape v 3.8.0 and StringApp confidence score cut-off 0.40). A green−black−red gradient was applied according to the iTRAQ ratio mean of down-regulated and up-regulated proteins. The circadian layout of the network is based on betweenness centrality. The diameters of nodes are proportional to betweenness centrality. (**D**) Gene ontology analysis deciphering the most significant biological processes with a lower *p* value of 0.05 in which the 29 secreted deregulated proteins are involved (in red for biological processes related to up-regulated proteins and in green for biological processes related to down-regulated proteins) (David Bioinformatics Resources v 6.8) (Raw data Appendix A).

**Figure 2 ijms-22-04316-f002:**
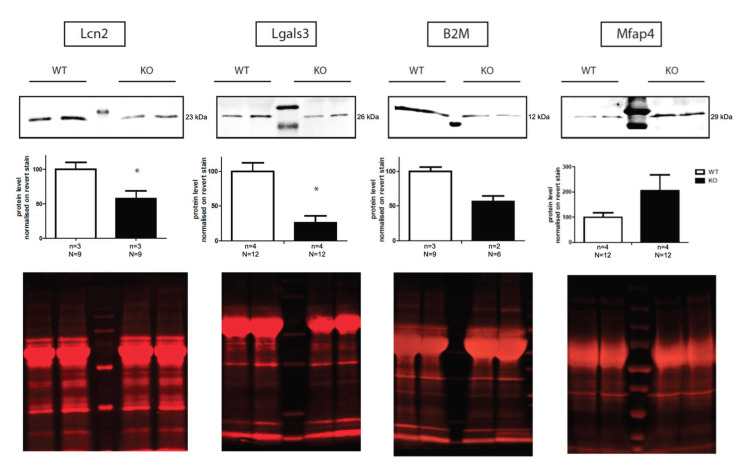
Western blot validation of proteomic studies of the glial secretome. Protein levels were quantified by western blot of concentrated conditioned-medium from WT and *Mecp2* KO astroglial culture. Each lane was a mix of conditioned-medium from 3 WT or 3 *Mecp2* KO pooled (*n* = number of experiments, N = number of animals). Representative western blot images are shown above each graph and the middle lane is the prestained protein ladder. Total protein staining normalization was used for quantification. Lcn2 and Lgals3 levels were significantly reduced in *Mecp2* KO glial secretome samples. Mfap4 and B2M were not found significantly altered in *Mecp2* KO samples compared to WT. Data are represented as mean ± SEM. Mann-Whitney rank sum test, * *p* < 0.05.

**Figure 3 ijms-22-04316-f003:**
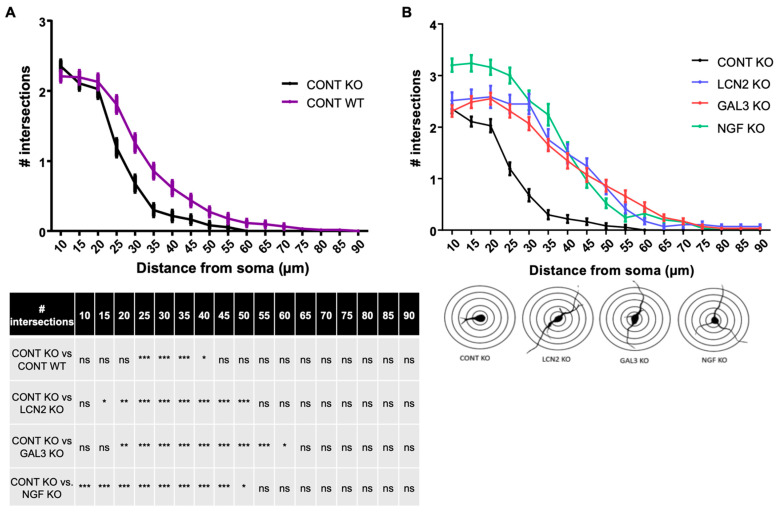
Sholl analysis of *Mecp2* KO cortical neurons after specific treatments. (**A**) *Mecp2* KO neurons show reduced dendritic arborization compared to the WT ones. (**B**) Lcn2, Lgals3 and NGF treatments increase number of intersections in *Mecp2* KO neurons. Significant differences between conditions and genotypes are summarized in tables. Data are presented as mean ± SEM. Two-way RM Anova followed by Turkey’s multiple comparisons test, * *p* < 0.05, ** *p* < 0.01, *** *p* < 0.001, ns: non-significant.

**Table 1 ijms-22-04316-t001:** Refined list of 29 secreted proteins significantly dysregulated in glial secretome of mouse Rett syndrome (iTRAQ labelled quantitative proteomic data obtained independently from 2 independent replicates with, in each experiment, 3 pooled two-day-old male mice for each condition (*Mecp2* KO and wild type)). Down- and up- regulated proteins are respectively in green and red in the table.

Accession	Name	Gene Name	Peptides (>95%)	Mean of iTRAQ Ratio KO/WT	*p* Value Rat. KO/WT
**P16045**	Galectin-1	Lgals1	9	0.14	1.73 × 10^−2^
**P11672**	Neutrophil Gelatinase-Associated Lipocalin	Lcn2	93	0.25	1.12 × 10^−5^
**P10148**	C-C Motif Chemokine 2	Ccl2	3	0.35	8.05 × 10^−4^
**P16110**	Galectin-3	Lgals3	6	0.35	1.10 × 10^−2^
**P24452**	Macrophage-Capping Protein	Capg	6	0.41	2.06 × 10^−4^
**Q61362**	Chitinase-3-Like Protein 1	Chi3l1	100	0.46	1.68 × 10^−6^
**P21460**	Cystatin-C	Cst3	69	0.48	4.76 × 10^−2^
**P01887**	Beta-2-Microglobulin	B2m	24	0.52	4.01 × 10^−4^
**Q9CXI5**	Mesencephalic Astrocyte-Derived Neurotrophic Factor	Manf	2	0.54	7.21 × 10^−4^
**Q8VCT3**	Aminopeptidase B	Rnpep	7	0.59	5.69 × 10^−3^
**Q91WP6**	Serine Protease Inhibitor A3N	Serpina3n	25	0.60	1.71 × 10^−2^
**Q06890**	Clusterin	Clu	92	1.33	8.28 × 10^−3^
**P01027**	Complement C3	C3	766	1.46	1.9 × 10^−2^
**Q91ZJ9**	Hyaluronidase-1	Hyal1	3	1.56	1.46 × 10^−3^
**Q9CQ01**	Ribonuclease T2	Rnaset2	15	1.62	1.03 × 10^−3^
**P01029**	Complement C4-B	C4b C4	141	1.85	1.86 × 10^−3^
**O88998**	Noelin	Olfm1	6	1.89	1.57 × 10^−3^
**Q61220**	Protein Kinase C-Binding Protein NELL2	Nell2	3	1.90	1.58 × 10^−3^
**O35684**	Neuroserpin	Serpini1	4	2.03	1.38 × 10^−4^
**P97298**	Pigment Epithelium-Derived Factor	Serpinf1	24	2.04	4.77 × 10^−3^
**B9EKR1**	Receptor-Type Tyrosine-Protein Phosphatase Zeta	Ptprz1	13	2.06	6.80 × 10^−3^
**O54782**	Epididymis-Specific Alpha-Mannosidase	Man2b2	46	2.11	2.72 × 10^−5^
**Q07797**	Galectin-3-Binding Protein	Lgals3bp	26	2.11	8.30 × 10^−4^
**O09114**	Prostaglandin-H2 D-Isomerase	Ptgds	22	2.17	2.02 × 10^−3^
**P50608**	Fibromodulin	Fmod	19	2.37	1.86 × 10^−4^
**Q99LS0**	Augurin	Ecrg4	8	2.53	2.49 × 10^−6^
**Q9D2L1**	Arylsulfatase K	Arsk	5	2.68	1.78 × 10^−2^
**Q9D1H9**	Microfibril-Associated Glycoprotein 4	Mfap4	2	2.75	4.80 × 10^−4^
**Q9EQG7**	Ectonucleotide Pyrophosphatase/Phosphodiesterase Family Member 5	Enpp5	13	2.88	3.05 × 10^−7^

## Data Availability

The secretomic data were deposited to the ProteomeXchange Consortium via the MassIVE Dataset Submission [24] under the MassIVE MSV000085665 identifier (http://massive.ucsd.edu) and ProteomeXchange PXD020143 identifier (http://www.proteomexchange.org).

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
