# Peer review of "Analysis of Astroglial Secretomic Profile in the Mecp2-Deficient Male Mouse Model of Rett Syndrome"

_ijms, 2021, doi:10.3390/ijms22094316_

Round 1

Reviewer 1 Report

Thank you for allowing me to review the manuscript by Ehinger and colleagues “Analysis of astroglial secretomic profile in the Mecp2-deficient male mouse model of Rett syndrome” an interesting study discovering potential astroglial secreted proteins involved in Rett neurophenotype that might represent actionable targets. 

The study design is well-described, achieving interesting conclusions.  My impression is positive although some clarifications might be warranted.

Regarding the proteomic analysis I would suggest to better explain the methods used for deregulated proteins’ list refinement. How could you exclude 76 proteins?

Minor

Results 2.1: One hundred and five proteins, where 105 was used elsewhere. I suggest to use 105.

Results 2.2: LCN2 and Lgals3. Please for consistency do not use capital letters for LCN2.

Please clarify the legend of Table 1. “iTRAQ labelled quantitative proteomic data obtained independently Figure 2. independent replicates with, in each ex- periment, 3 pooled two-day-old male mice for each condition (Mecp2-KO and wild type”

In Results 2.3 I suggest to point out the results obtained by Western Blot vs Tandem Mass. I suppose results from Western blot correspond to LCN2 (-42%), Lgals3 (-44%) and B2M (-43,4%), might you specify this?

2.6 “Seven PND male pups” might you clarify the postnatal day of these pups?

Author Response

Dear editor,

Please find enclosed a revised version of our research article submitted the 5th of March for the special issue of the International Journal of Molecular Sciences entitled “Molecular Research on Rett Syndrome and Related Disorders”

We hope that the revised manuscript entitled “Astroglial secretome analysis of a mouse model of Rett syndrome: pathological and therapeutic implications” This revised version takes into account and answers the comments that were made by the two previous reviewers.

Reviewer 1

Comments and Suggestions for Authors

Thank you for allowing me to review the manuscript by Ehinger and colleagues “Analysis of astroglial secretomic profile in the Mecp2-deficient male mouse model of Rett syndrome” an interesting study discovering potential astroglial secreted proteins involved in Rett neurophenotype that might represent actionable targets.

The study design is well-described, achieving interesting conclusions. My impression is positive although some clarifications might be warranted.

Regarding the proteomic analysis I would suggest to better explain the methods used for deregulated proteins’ list refinement. How could you exclude 76 proteins?

We thank the reviewer to give us the opportunity for clarifying the refinement process of the secreted protein list. In addition to a Gene Ontology (GO) annotation filtering with the terms related to secreted proteins (extracellular exosome, extracellular space and extracellular region), we used the Uniprot annotation of protein subcellular location that contains both expertly curated data from literature and rule-based automatic annotation. We are aware that this refinement is pretty stringent, but we prefer to present network of 29 proteins recorded as secreted in both Uniprot annotation and GO term rather than presenting a larger number of proteins for which the secretion status is less confident.

To clarify this point in the manuscript, we remplaced the following sentences in page 3.

" The functional analysis performed using David Bioinformatics resources (https://david.ncifcrf.gov/summary.jsp) confirmed that the quantified proteins were mostly localized in extracellular compartments (extracellular exosome, extracellular space and extracellular region associated to a P-value of 2.7x10-47, 2.9x10-15 and 3.1x10-13 , respectively) (Table S2b). For more specificity in the description of the mice astroglial secretome changes in the context of Mecp2 deficiency, we refined the list of these 105 deregulated proteins to proteins annotated as secreted proteins in the UniProtKB/Swissprot database and identified 29 secreted proteins to be significantly deregulated in the RTT astroglial secretome (Table 1 and Table S3a)."

by

" The functional analysis performed using David Bioinformatics resources (https://david.ncifcrf.gov/summary.jsp) confirmed that the quantified proteins were mostly localized in extracellular compartments (extracellular exosome, extracellular space and extracellular region associated to a P-value of 2.7x10-47, 2.9x10-15 and 3.1x10-13 , respectively) (Table S2b). In addition to these Gene Ontology (GO) annotations, the Uniprot annotation provides subcellular localization of proteins with both expertly curated data from literature and rule-based automatic annotation. Therefore, we further filtered the list of our deregulated proteins by using the subcellular localisation of Uniprot annotation. Altogether, by considering GO terms and Uniprot annotations, twenty-nine secreted deregulated proteins were retained (Table 1 and Table S3a) to build a network of secreted deregulated proteins upon Mecp2 deficiency in the RTT astroglial secretome (Figure 1C)."

We also modified the table S3a (Refined list of 29 secreted proteins signicantly deregulated (DEP) in glial secretome of mouse Rett syndrome) where the uniprot annnotation was already indicated by adding the information on the GO term annotation for extracellular exosome, extracellular space and extracellular region.

Minor

Results 2.1: One hundred and five proteins, where 105 was used elsewhere. I suggest to use 105.

This has been corrected

Results 2.2: LCN2 and Lgals3. Please for consistency do not use capital letters for LCN2.

This has been corrected

Please clarify the legend of Table 1. “iTRAQ labelled quantitative proteomic data obtained independently Figure 2. independent replicates with, in each ex- periment, 3 pooled two-day-old male mice for each condition (Mecp2-KO and wild type”

We edited the sentence as follows: “iTRAQ labelled quantitative proteomic data obtained independently. Mean of iTRAQ ratio KO/WT is the mean of the iTRAQ ratio for each protein from Mecp2-KO mice versus wild type mice. Two independent replicates with, in each experiment, 3 pooled two-day-old male mice for each condition (Mecp2-KO and wild type (WT))”

In Results 2.3 I suggest to point out the results obtained by Western Blot vs Tandem Mass. I suppose results from Western blot correspond to LCN2 (-42%), Lgals3 (-44%) and B2M (-43,4%), might you specify this?

This has been specified

2.6 “Seven PND male pups” might you clarify the postnatal day of these pups?

The sentence has been clarified “On PND 7, male pups were rapidly decapitated, and the cortices were dissected”.

Reviewer 2

Comments and Suggestions for Authors

Analysis of astroglial secretomic profile in the Mecp2-deficient male mouse model of Rett syndrome

by Y. Ehinger et al.

The authors present a secretome study focussing on Rett syndrome. This adds to extant data, the list of deregulated proteins provided represents an advance in itslef, it is highly informative and may inspire future studies on the study of pathomechanisms. It is technically fine and well-written.

I feel there is a major limitation of the study as presented, which should be frankly stated in the text. Also, the conclusions of the experiment in cultured neurons may not be justified.

MAJOR POINTS

It is not clear that the secretome analysis is based on astrocytes, rather a mixed culture of astrocytes and microglia.

Changes and clarifications have been made to answer the reviewer’s concern.

- Cell culture: Include in “Limitations” paragraph that culture has not been assayed for astrocyte enrichment (e.g. by % GFAP+ cells).

This has been clarified in the limitations.

- Also, what would be the rationale to assume secretion of Lgals3 from astrocytes, when most Lgals3 data published are related to microglia. This further questions the astroglial nature of the secretome analysed.

Obtaining pure astrocyte cultures from brains is something very complex to achieve and in our case, we found that our cultures were very weakly contaminated by microglial cells. Although the microglial cells were in very low numbers, we can't totally claim that our results were obtained from astrocytic cells only. However, it was already showed that Lgals3 is expressed by astrocytes (Sirko et al, 2015).

- The entire paper, including Title should reflect this, and not convey the impression that specifically astrocytes are analzed here.

We agree it is noteworthy our cultures are not pure astrocytes but a low percentage of microglial cells is present, this is why we refer to the cultures as “astroglial” in the title and changes were made throughout the text to clarify.

Para 2.4 states that “LCN2 and Lgals3 supplementation improves dendritic arborization of Mecp2 KO neurons”. This may be misleading since the effects of LCN2 and GAL3 on dendritic morphology are not really conclusive in the paper`s context.

Two factors (LCN2 and GAL3) isolated during analysis increase dendritic morphology in both, mutant and WT neurons; however, this is shown in separate plots for mutant and WT. The authors argue that reduced dendritic morphology is a hallmark of and implicated in the pathogenesis of Rett syndrome, but to begin with it is not clear whether dendrites of untreated mutant and WT neurons are significantly different. To me it looks as if this is not the case (maybe an issue of cell culture?), these WT and mutant Sholl values shoud be compared in a single plot. If in the WT one cannot speak of “improved” dendritic morphology by these factors (since it is normal by definition), but “increased”, all these data would show is that dendritic morphology is comparable in wt and mutant neurons and that the factors increase dendritic morphology in both.

This should be frankly communicated, as a negative finding not supporting the authors` hypothesis – in parapgrpahs in 2.4, - 3.4 (rewrite last two sentences)

In order to clarify our results, we have added a figure comparing WT and KO neurons to confirm that KO neurons appear less tree-like than those of WT mice (Figure 3A).  

Some differences detected in the secretomes relate to proteins differentially regulated only at the level of secretion, but not translation (2.3) – how could this be without proteins accumulating intracellularly? Others are regulated at the level of transcription (Lcn2, [17]). Please comment on and discuss this.

We agree with the reviewer that a secretion deficit should be linked with a protein accumulation, however we did not find any difference between protein levels in Mecp2-KO and WT astroglial cultures. This deficit may be due to other mechanisms. Moreover, Mecp2 is a very complex protein because it is a transcription factor, a regulator of chromatid architecture and a splicing regulator. It has an impact on the regulation of a very large number of genes resulting in a biological complexity that is difficult to assess.

EDITING

- Fig S4 or S5 missing, incorrectly labelled

Both figures are present in the pdf of suppl figures and cited in paragraphs 2.4 and 3.1 respectively.

- Arrangement and/ or naming of some Figures and Tables is inadequate. – Arrange Figs. as numbered, also check text and Suppl Data for Fig numbering. The Excel tables should be included in Suppl. Data, and not go as unpublished. Some Suppl. Tables have confusing, inconsistent numbering, regarding Title vs. spreadsheet name.

Places of Figure S2 and Figure S3 have been exchanged to correct the arrangement.

Titles and legends of suppl. tables have been corrected to be consistent with the text.

- Repeated introduction of abbreviations

Abbreviations have been introduced both in the abstract and in the introduction for ease of reading.

We hope that our updated manuscript will satisfy the reviewers and fulfil the publishing criteria of the International Journal of Molecular Sciences.

We therefore hope that you will find our new manuscript suitable for further consideration.

Yours sincerely.

Jean-Christophe Roux, PhD

Reviewer 2 Report

Analysis of astroglial secretomic profile in the Mecp2-deficient male mouse model of Rett syndrome

by Y. Ehinger et al.

The authors present a secretome study focussing on Rett syndrome. This adds to extant data, the list of deregulated proteins provided represents an advance in itslef, it is highly informative and may inspire future studies on the study of pathomechanisms. It is technically fine and well-written.

I feel there is a major limitation of the study as presented, which should be frankly stated in the text. Also, the conclusions of the experiment in cultured neurons may not be justified.

MAJOR POINTS

It is not clear that the secretome analysis is based on astrocytes, rather a mixed culture of astrocytes and microglia.

- Cell culture: Include in “Limitations” paragraph that culture has not been assayed for astrocyte enrichment (e.g. by % GFAP+ cells).

- Also, what would be the rationale to assume secretion of Lgals3 from astrocytes, when most Lgals3 data published are related to microglia. This further questions the astroglial nature of the secretome analysed.

- The entire paper, including Title should reflect this, and not convey the impression that specifically astrocytes are analzed here.

Para 2.4 states that “LCN2 and Lgals3 supplementation improves dendritic arborization of Mecp2 KO neurons”. This may be misleading since the effects of LCN2 and GAL3 on dendritic morphology are not really conclusive in the paper`s context. 
Two factors (LCN2 and GAL3) isolated during analysis increase dendritic morphology in both, mutant and WT neurons; however, this is shown in separate plots for mutant and WT. The authors argue that reduced dendritic morphology is a hallmark of and implicated in the pathogenesis of Rett syndrome, but to begin with it is not clear whether dendrites of untreated mutant and WT neurons are significantly different. To me it looks as if this is not the case (maybe an issue of cell culture?), these WT and mutant Sholl values shoud be compared in a single plot. If in the WT one cannot speak of “improved” dendritic morphology by these factors (since it is normal by definition), but “increased”, all these data would show is that dendritic morphology is comparable in wt and mutant neurons and that the factors increase dendritic morphology in both.
This should be frankly communicated, as a negative finding not supporting the authors` hypothesis – in parapgrpahs in 2.4, - 3.4 (rewrite last two sentences)

Some differences detected in the secretomes relate to proteins differentially regulated only at the level of secretion, but not translation (2.3) – how could this be without proteins accumulating intracellularly? Others are regulated at the level of transcription (Lcn2, [17]). Please comment on and discuss this.

EDITING

- Fig S4 or S5 missing, incorrectly labelled

- Arrangement and/ or naming of some Figures and Tables is inadequate. – Arrange Figs. as numbered, also check text and Suppl Data for Fig numbering. The Excel tables should be included in Suppl. Data, and not go as unpublished. Some Suppl. Tables have confusing, inconsistent numbering, regarding Title vs. spreadsheet name.

- Repeated introduction of abbreviations

Author Response

(The authors gave the same response as above.)
